

# Diverse responses of *Symbiodinium* types to menthol and DCMU treatment

Jih-Terng Wang[1], Shashank Keshavmurthy[2], Tzu-Ying Chu[1] and Chaolun Allen Chen[2,3]

[1] Graduate Institute of Biotechnology, Tajen University, Pingtung, Taiwan
[2] Biodiversity Research Center, Academia Sinica, Taipei, Taiwan
[3] Institute of Oceanography, National Taiwan University, Taipei, Taiwan

## ABSTRACT

To understand the mechanism of photosynthetic inhibition and generation of reactive oxygen species (ROS) in *Symbiodinium* types under stress, chemicals such as dichlorophenyl dimethylurea (DCMU) are widely used. Moreover, DCMU and recently menthol were used to generate aposymbiotic cnidarian hosts. While the effects of DCMU on *Symbiodinium* cells have been extensively studied, no studies have shown the mechanism behind menthol-induced coral bleaching. Moreover, no study has compared the effects of DCMU and menthol treatments on photosystem II (PSII) activity and generation of ROS in different *Symbiodinium* types. In this study, we utilized five freshly isolated *Symbiodinium* types (*S. minutum* (B1), *S. goreaui* (C1), C3, C15, and *S. trenchii* (D1a)) to compare the effects of DCMU and menthol treatments. *Symbiodinium* cells were exposed to DCMU and menthol at different concentrations for 4 h. Results showed that values of the 50% inhibitory concentration ($IC_{50}$) for PSII inhibition were 0.72∼1.96 mM for menthol-treated cells compared to 29∼74 pM for DCMU-treated cells. Diverse responses of *Symbiodinium* types were displayed in terms of PSII tolerance to menthol (*S. minutum* > *S. trenchii* = C15 > C3 = *S. goreaui*), and also in the response curves. In contrast, responses were not so diverse when the different types were treated with DCMU. Three of five menthol-treated *Symbiodinium* types showed instant and significant ROS generation when PSII activity was inhibited, compared to no ROS being generated in DCMU-treated *Symbiodinium* types. Both results indicated that menthol inhibited *Symbiodinium* PSII activity through *Symbiodinium* type-dependent mechanisms, which were also distinct from those with DCMU treatment. This study further confirmed that photosynthetic functions *Symbiodinium* have diverse responses to stress even within the same clade.

# INTRODUCTION

*Symbiodinium* spp. are associated with marine invertebrate hosts, including Protista, Porifera, Cnidaria, and Mollusca (*Coffroth & Santos, 2005*), and play important functional roles in providing photosynthesis-derived carbon and conserving or recycling host nitrogen metabolites (*Davy, Allemand & Weis, 2012*). To date, nine (A∼I) *Symbiodinium* clades and numerous subcladal types have been identified with distinguishable genetic identities and

Corresponding authors
Jih-Terng Wang,
jtwtaiwan@gmail.com,
jtw@tajen.edu.tw
Chaolun Allen Chen,
cac@gate.sinica.edu.tw

physiological characteristics (reviewed in *Stat & Gates, 2011*). Among these, only clades A, B, C, and D are widely associated with scleractinian corals. Different *Symbiodinium* types are known to show variations in their photosynthesis functions (*Rowan, 2004*; *Tchernov et al., 2004*; *Robinson & Warner, 2006*; *Sampayo et al., 2008*; *Wang et al., 2012b*; *Suggett et al., 2015*), resulting in different contributions to their symbiotic associations (*Stat, Morris & Gates, 2008*; *Yuyama & Higuchi, 2014*; *Pernice et al., 2015*). Various stress tolerabilities among different *Symbiodinium* types were also revealed by high antioxidant plasticity (*Krueger et al., 2014*).

The association between corals and *Symbiodinium* is highly vulnerable to physical and chemical disturbances. For example, only 1~2 °C above the summer average under moderate to high irradiance is enough to disrupt the symbiotic relationship and end up with loss of symbionts from coral hosts, resulting in 'coral bleaching' (*Fitt et al., 2001*; *Lesser & Farrell, 2004*). The mechanism of *Symbiodinium* depletion (reviewed in *Weis, 2008*), from cnidarian host cells, is generally attributed to damage by reactive oxygen species (ROS) generated from photoinhibition of *Symbiodinium* during stress (e.g., high temperature, irradiance, or herbicides, e.g., dichlorophenyl dimethylurea (DCMU) treatment) (*Jones et al., 1998*; *Jones, 2004*; *Jones, 2005*) or stress on the coral host's metabolism (*Jones, 2004*). Disruption of *Symbiodinium*-coral symbiosis can also be achieved by some chemicals such as heavy metals (*Jones, 1997* and sunscreen contamination (*Danovaro et al., 2008*). However, there is no direct evidence to indicate coral bleaching caused by ROS generated by DCMU-treated *Symbiodinium* (*Jones, 2004*).

While use of DCMU to generate aposymbiotic cnidarian hosts and the effects of DCMU on *Symbiodinium* cells have extensively been studied (*Murata et al., 2007*; *Takahashi et al., 2013*; *Fransolet, Roverty & Plumier, 2014*; *Aihara, Takahashi & Minagawa, 2016*; *Parrin & Blackstone, 2017*), use of menthol as an efficient bleaching agent has only recently been established. In recent years, menthol was found to successfully induce bleaching in coral and sea anemone hosts (*Wang et al., 2012a*; *Dani et al., 2016*; *Matthews et al., 2016*).

Menthol is a cyclic terpene alcohol which can cause local anesthetic effects in neuronal and skeletal muscles by blocking voltage-operated sodium channels (*Haeseler et al., 2002*). This effect has led to its use as a marine anesthetic (*Moore, 1989*; *Lauretta et al., 2014*). A variety of different membrane receptors are known to respond to menthol stimulation, including transient receptor potential (TRP)M8 that results in an increase in intracellular $Ca^{2+}$ concentrations and causes a cold sensation in vertebrates (*McKemy, Neuhausser & Julius, 2002*; *Okazawa et al., 2000*; *Peier et al., 2002*; *Hans, Wilhelm & Swandulla, 2012*). Menthol is also a cytotoxic compound to plant tissues (*Muller & Hauge, 1967*; *Brown, Hegarty & Charlwood, 1987*), causing a drastic reduction in the number of intact mitochondria and Golgi bodies in seedling roots (*Lorber & Muller, 1976*), inhibiting respiration and photosynthesis (*Pauly, Douce & Carde, 1981*), and decreasing cell membrane permeability (*Muller et al., 1969*).

Recently, *Wang et al. (2012a)* demonstrated that menthol can be used to effectively to induce bleaching in the corals *Isopora palifera* and *Stylophora pistillata*. *Dani et al. (2016)* further hypothesized that menthol-induced activation of a host TRP receptor might increase the intracellular calcium concentration. Modulation of calcium homeostasis,

which is known to upregulate the autophagic pathway (*Smaili et al., 2013*), could therefore trigger *Symbiodinium* depletion through a phagolysosomal process. Whether the PSII breakdown in the endosymbiotic *Symbiodinium* in corals is directly or indirectly caused by menthol needs to be clarified. Moreover, no studies have compared responses of different *Symbiodinium* types to both menthol and DCMU, which would investigate the important but under-explored topic of functional diversity among *Symbiodinium* species.

In this study, we tested the responses of different *Symbiodinium* types to DCMU and menthol exposure by evaluating the concentration for 50% inhibition of PSII activity and the generation of ROS. Results of this study will provide information on diverse responses among different *Symbiodinium* types to menthol treatment and is a first step towards understanding mechanisms of *Symbiodinium* cell depletion caused by menthol treatment.

## MATERIALS AND METHODS

*Symbiodinium* were isolated from five coral species. Parts of colonies of *Acropora humilis*, *Galaxea fasicularis*, *Isopora palifera,* and *Porites lutea* corals were collected from reefs within Kenting National Park, Taiwan (21°55′54″N, 120°44′45″E). A sample collection permit (no. 488-100-01) was obtained from the Kenting National Park Authority as part of the long-term environmental monitoring project. Coral colonies were transferred to the laboratory within 3 h in an aerated plastic box, and maintained in an aquarium tank under conditions described in *Wang et al. (2011)*. Corals were acclimatized to laboratory aquarium (*Wang et al., 2012a*; *Wang et al., 2012b*) settings (25 °C and 70 μmol photons $m^{-2}$ $s^{-1}$ light) for 1 week before initiating experiments. The glass sea anemone, *Exaiptasia pulchella* (*Grajales & Rodríguez, 2014*), originally collected from a discharge trench of the Tongkun Fishery Research Institute, Taiwan, had been maintained in a tank ($45 \times 30 \times 30$ cm) equipped with illumination and temperature control for more than 1 year.

To prepare freshly isolated *Symbiodinium* (FIS), coral fragments with 5∼10 $cm^2$ of live tissue and excised tentacles from sea anemones were homogenized and washed with artificial seawater (ASW) prepared from Instant Ocean (Aquarium Systems, France) as described by *Wang et al. (2011)*. The dominant FIS types used in this study were identified as *Symbiodinium* ITS2 types C1 (*S. goreaui*), C3, C15, D1a (*S. trenchii*), and B1 (*S. minutum*) from *S. pistillata*, *A. humilis*, *P. lutea*, *G. fasicularis* and *E. pulchella*, respectively, based on denaturation gradient gel electrophoresis (DGGE) according to *Wang et al. (2011)*.

PSII activities of FIS were determined by the maximum quantum yield ($F_v/F_m$), and the minimum ($F_0$) and maximum ($F_m$) fluorescence levels were measured to calculate the variable fluorescence [$F_v$, ($F_v = F_m - F_0$)] using a DIVING-PAM fluorometer (Walz, Germany) at a setting of 8 for the measuring light and saturating flash of actinic light.

When determining values of the half-maximal inhibitory concentration ($IC_{50}$), triplicate samples with 5 ml of FIS (around $10^6$ cells $ml^{-1}$) of each type showing $F_v/F_m$ values of >0.6 (a value generally considered normal/healthy, *Fitt et al., 2001*) were transferred to a 50-ml Falcon tube and centrifuged at $417 \times$ g for 3 min to collect *Symbiodinium*. The algal pellet was then re-suspended in 5 ml of different concentrations of menthol- or DCMU-ASW and maintained at 25 °C under illumination of around 70 μmol photons $m^{-2}$ $s^{-1}$ of

photosynthetically active radiation (PAR). *Wang et al. (2011)* suggested that the PSII activity of FIS begins to fluctuate after incubation for 4 h, so the effects of menthol and DCMU were determined as the 4-h $IC_{50}$. $F_v/F_m$ values of FIS in menthol or DCMU were directly determined with the DIVING-PAM fluorometer without dark adaptation at 4 h of incubation, and were converted to PSII inhibition by comparing to values in fresh isolates as below:

PSII inhibition (%) $= (1 - ((F_v/F_m)_{4\,h}/(F_v/F_m))_{0\,h}) \times 100\%$;

where $(F_v/F_m)_{0\,h}$ and $(F_v/F_m)_{4\,h}$ values were respectively measured at 0 and after 4 h of incubation. After plotting PSII inhibition against the logarithm-transformed concentration of menthol (or DCMU), the equation for calculating the $IC_{50}$ was obtained from the best curve-fit-model provided in SigmaPlot 10.0 software.

ROS generated in menthol- or DCMU-treated FIS were determined with a $2',7'$-dichlorofluorescin diacetate ($H_2$DCFDA) probe. FIS (around $10^6$ cells ml$^{-1}$, cells counted as in *Wang et al., 2012b*) of each *Symbiodinium* type was incubated in ASW containing 1.73 mM menthol (for *S. minutum*, this was 2.43 mM) or 130 pM DCMU as the positive control, for 4 h in which the PSII activity of FIS would be completely inhibited. The mentioned concentration used to examine ROS generation refers to the highest concentration causing maximum inhibition of PSII activity. Also, during the trials of menthol-induced bleaching of the coral hosts, the effective concentration varied between different hosts associated with different *Symbiodinium* types, but all displayed significant photosystem II (PSII) breakdown in the endosymbiotic *Symbiodinium* (Fig. S1). ROS in *Symbiodinium* were detected by incubating 1 ml of an algal suspension with 5 µl $H_2$DCFDA (10 mM in dimethyl sulfoxide) for 30 min in the dark, followed by fluorescence microscopic examination (*Mydlarz & Jacobs, 2004*) and fluorescence determination (*Wang et al., 2011*). The fluorescence data of ROS signals are expressed in arbitrary fluorescence units per cell.

Dose effects (4-h $IC_{50}$) of menthol and DCMU on PSII inhibition of FIS were calculated from samples of each treatment. The curve equations for $IC_{50}$ calculation were determined by the best-fit regression curve of PSII inhibition (%) against the logarithm-transformed reagent concentration, which are described in Eq. (1) for menthol and (2) for DCMU treatment:

$$y = y_0 + \frac{a}{1 + e^{\frac{-(x-x_0)}{b}}} \tag{1}$$

and

$$y = \frac{a}{1 + e^{\frac{-(x-x_0)}{b}}}. \tag{2}$$

Parameters and regression coefficients of the equations are listed in Table 1.

Comparisons of 4-h $IC_{50}$ values between *Symbiodinium* types were made using a one-way analysis of variance (ANOVA) followed by Fisher's least significance difference (LSD) test, with a significance level of 0.05. The coefficient of variance (CV) was used to evaluate the variation in $IC_{50}$ measurements between replicates.

**Table 1** Parameters and regression coefficients of equations for dose–response curves derived from menthol and dichlorophenyl dimethylurea (DCMU) (Eqs. (1) and (2) in 'Material and Methods') treatments on photosystem II (PSII) inhibition of different types of freshly isolated *Symbiodinium*. Data are presented as mean values ($n = 3$).

| Phylotype | Menthol | | | | | DCMU | | | |
|---|---|---|---|---|---|---|---|---|---|
| | $a$ | $b$ | $x_0$ | $y_0$ | $r^2$ | $a$ | $b$ | $x_0$ | $r^2$ |
| B1 (*S. minutum*) | 53.71 | 0.045 | 0.23 | 6.33 | 0.992 | 86.57 | 0.18 | 2.06 | 0.997 |
| C1 (*S. goreaui*) | 98.35 | 0.115 | −0.11 | 8.13 | 0.995 | 72.86 | 0.21 | 1.84 | 0.997 |
| C3 | 94.45 | 0.050 | −0.06 | 6.44 | 0.996 | 74.39 | 0.23 | 1.59 | 0.999 |
| C15 | 96.63 | 0.033 | 0.01 | 2.66 | 0.999 | 75.73 | 0.16 | 1.86 | 0.995 |
| D1a (*S. trenchii*) | 111.22 | 0.105 | 0.07 | 10.15 | 0.989 | 73.53 | 0.22 | 1.57 | 0.998 |

## RESULTS

When *Symbiodinium* algae were incubated in menthol-supplemented ASW for 4 h, all five *Symbiodinium* types (*S. minutum*, *S. goreaui*, C3, C15, and *S. trenchii*) displayed typical dose–response curves under menthol concentrations of 0.19∼2.43 mM (Fig. 1A). In contrast to menthol inhibition at millimolar levels, DCMU-treated samples displayed PSII inhibition of FIS at picomolar levels (4∼129 pM) (Fig. 1B).

Regression coefficients for curve fitting ranged 0.989∼0.999 for menthol treatments and 0.995∼0.999 for DCMU treatments. When converting parameter "$b$" in Table 1 to a slope factor (curve steepness) as described by *Motulsky & Christopoulos (2003)*, values for menthol treatments were divided into two groups which included high (20.0∼30.3 for *S. minutum*, types C3, and C15) and low slope factors (8.7∼9.5 for *S. goreaui* and *S. trenchii*). But those for DCMU-treated FIS displayed only one group of slope factors that ranged 4.3∼6.3. The 4-h $IC_{50}$ values for menthol and DCMU treatments on five *Symbiodinium* types were calculated, and results are listed in Table 2. Mean 4-h $IC_{50}$ values for menthol ranged 0.72∼1.96 mM with CV values ranging 1.1%∼12.3%, and those for DCMU ranged 29∼74 pM with CV values of 2.3%∼14.1% (Table 2). Two sets of data significantly varied among *Symbiodinium* types (menthol, $F_{4,11} = 17.54$, $p < 0.001$; DCMU, $F_{4,10} = 12.00$, $p < 0.01$), and *S. minutum* was found to be the most tolerant type to both menthol and DCMU irritation ($p < 0.05$). In contrast to the comparable tolerability to DCMU among different *Symbiodinium* types, *S. trenchii* and type C15 were significantly more tolerant to menthol than were *S. goreaui* and C3 ($p < 0.05$).

In order to explore if ROS are the cause of menthol inhibition of PSII activity of FIS, the generation of algal ROS under a condition of complete PSII activity inhibition by menthol or DCMU treatment was further examined. According to a fluorescence microscopic examination, no DCMU-treated type displayed green fluorescence, indicating no ROS signal, as found in the ASW control (Fig. S2). However, ROS generation in menthol-treated *Symbiodinium* varied among types. As shown in Fig. 2B, *S. minutum*, consistent with the ASW control in Fig. 2A, showed almost no ROS signal. For *S. trenchii*, there were only mild ROS signals detected (Fig. 2C). In contrast to *S, minutum* and *S. trenchii*, as shown in Figs. 2D–2F, types C15, C3, and *S. goreaui* displayed intense green fluorescence, representing considerable ROS generation. Direct measurements of the fluorescence

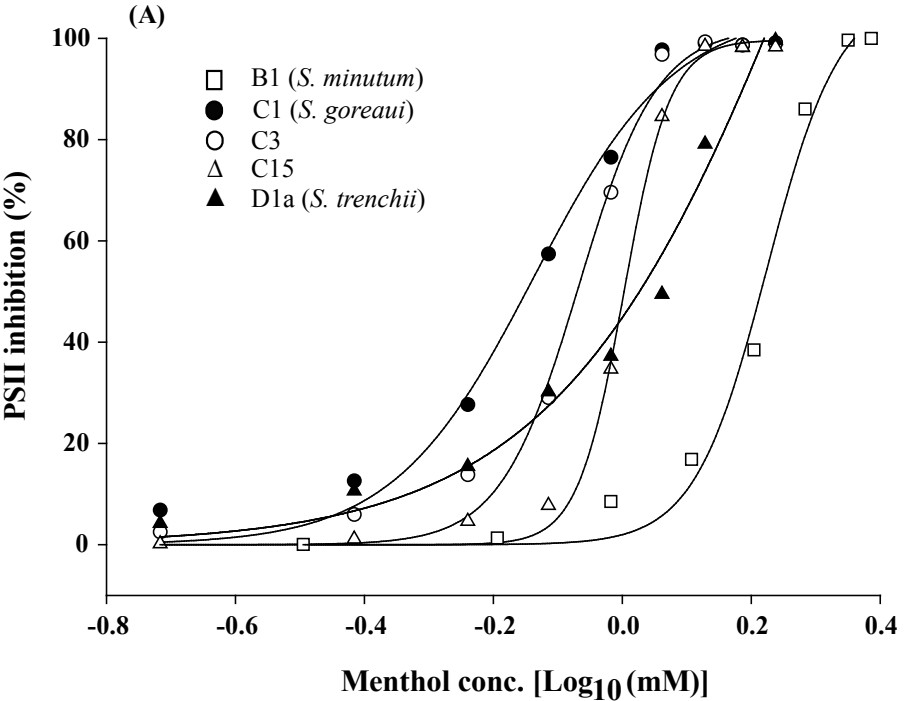

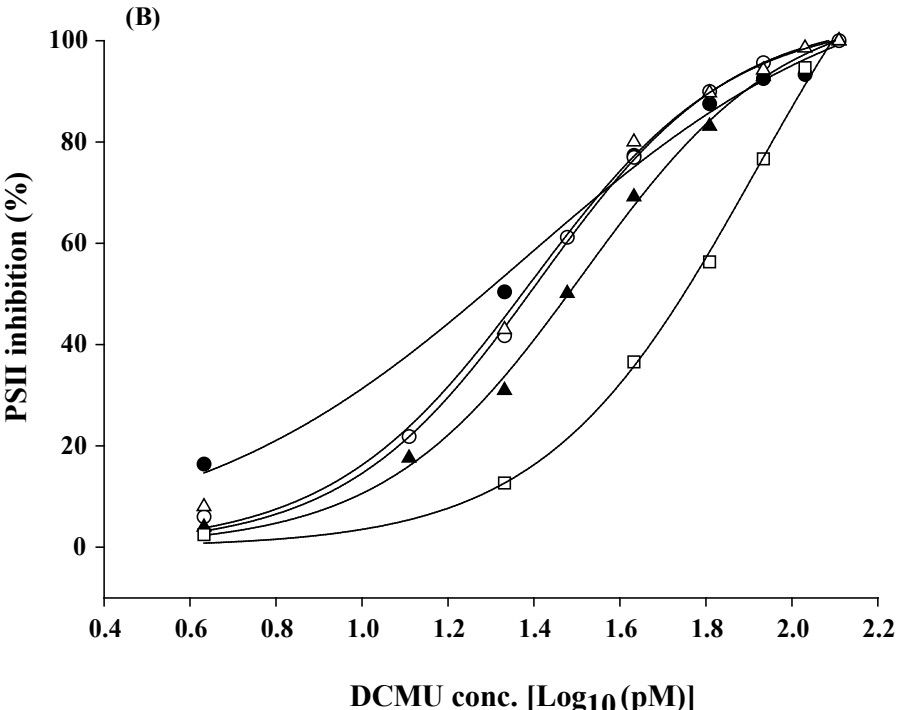

**Figure 1 Dose effects of menthol and dichlorophenyl dimethylurea (DCMU) on inactivation of photosystem II (PSII) function in freshly isolated *Symbiodinium*.** The means of PSII inhibition ($n = 3$) after 4 h of incubation in menthol (A) or DCMU (B) are plotted in the presence of various reagent concentrations.

**Table 2** **Values of 4-h 50% inhibitory concentration ($IC_{50}$) of menthol and dichlorophenyl dimethylurea (DCMU) to inactivate photosystem II (PSII) activity of freshly isolated *Symbiodinium* types.** Data are the mean $\pm$ SD ($n = 3$), and those with the same superscripts do not significantly differ at $p = 0.05$ (by Fisher's least significance difference test). CV indicates the coefficient of variation.

| Phylotype | Cnidarian host | Menthol | | DCMU | |
|---|---|---|---|---|---|
| | | mM | CV (%) | pM | CV (%) |
| B1 (*S. minutum*) | *Exaiptasia pulchella* | $1.96 \pm 0.13^a$ | 6.4 | $74 \pm 8^a$ | 10.9 |
| C1 (*S. goreaui*) | *Stylophora pistillata* | $0.72 \pm 0.09^b$ | 12.3 | $29 \pm 3^b$ | 9.8 |
| C3 | *Acropora humilis* | $0.86 \pm 0.02^b$ | 2.7 | $37 \pm 1^b$ | 2.3 |
| C15 | *Porites lutea* | $1.01 \pm 0.01^c$ | 1.1 | $33 \pm 3^b$ | 10.1 |
| D1a (*S. trenchii*) | *Galaxea fasicularis* | $1.07 \pm 0.07^c$ | 6.2 | $52 \pm 7^b$ | 14.1 |

Notes.
 a, b and c indicate the significance between data after statistical analysis at $P = 0.05$.

**Table 3** **Reactive oxygen species (ROS) signals detected in menthol- or dichlorophenyl dimethylurea (DCMU)-treated freshly isolated *Symbiodinium* (FIS) labeled with 2′, 7′-dichlorofluorescin diacetate.** Detailed conditions of menthol and DCMU treatment are described in Fig. 2, except that the concentration used for DCMU was 0.13 mM. ROS data are expressed as the fluorescence intensity of 2′, 7′-dichlorofluorescein, and are presented as the mean $\pm$ SD ($n = 3$).

| Phylotypes | ROS signal | |
|---|---|---|
| | Menthol | DCMU |
| | (fluorescence units cell$^{-1}$) | |
| B1 (*S. minutum*) | $18 \pm 2$ | $15 \pm 2$ |
| C1 (*S. goreaui*) | $64 \pm 6$ | $12 \pm 1$ |
| C3 | $68 \pm 9$ | $18 \pm 2$ |
| C15 | $58 \pm 8$ | $18 \pm 1$ |
| D1a (*S. trenchii*) | $35 \pm 3$ | $17 \pm 2$ |

intensity derived from an ROS probe also indicated that the five *Symbiodinium* types had varied responses to menthol irritation (Table 3). Menthol-treated *S. goreaui*, C3, and C15 types displayed almost double the extent of ROS fluorescence (with mean values of 58~68 fluorescence units cell$^{-1}$) of that of *S. trenchii* ($35 \pm 3$ fluorescence units cell$^{-1}$), and more than three times that of *S. minutum* ($18 \pm 2$ fluorescence units cell$^{-1}$) (Table 3). ROS fluorescence intensities obtained from menthol-treated *S. minutum* and all DCMU-treated types were comparable to background levels (with mean values of 18~28 fluorescence units cell$^{-1}$), even though $F_v/F_m$ values were reduced to <0.1. In order to monitor the times needed for *Symbiodinium* types to respond to menthol irritation, $F_v/F_m$ values and ROS levels of the C3 type were examined every 5 or 10 min after suspending the alga in menthol-supplemented ASW. As shown in Fig. 3A, the $F_v/F_m$ value was significantly reduced from 0.685 to <0.2 within 5 min of incubation, and to nearly 0 after 10 min of incubation. As with the decrease in $F_v/F_m$ value, ROS generation in the menthol treated *Symbiodinium* reached a maximum within 5 min of incubation (Fig. 3B).

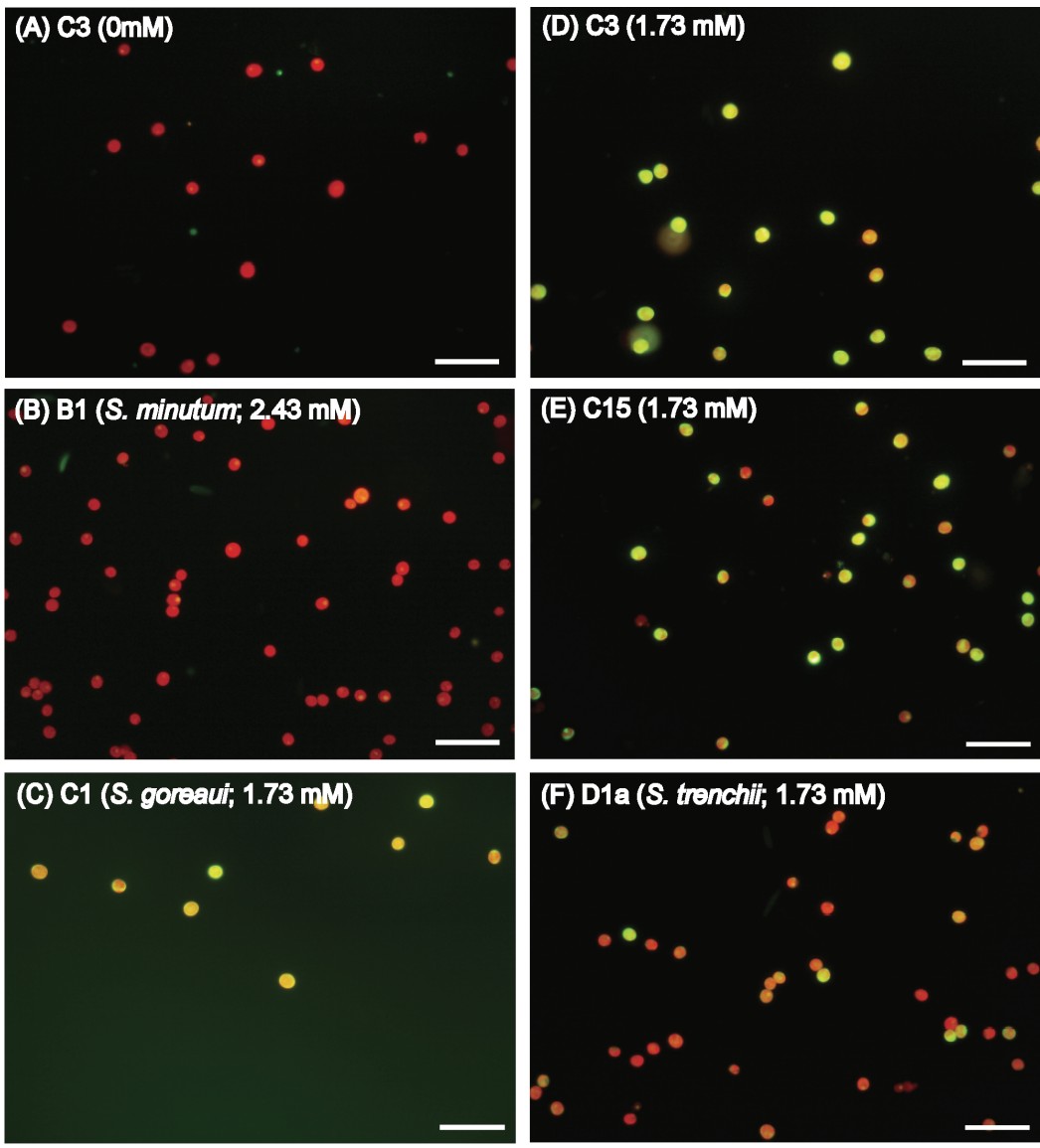

**Figure 2  Diverse reactive oxygen species (ROS) generation levels among *Symbiodinium* types when treated with artificial seawater (ASW) containing menthol.** Freshly isolated *Symbiodinium* (FIS) was incubated in menthol-supplemented ASW for 4 h, followed by 2′, 7′-dichlorofluorescin diacetate labeling and microscopic examination of fluorescence. Menthol concentrations used in FIS incubation were 1.73 mM for types C1 (*S. goreaui*), C3, C15, and D1a (*S. trenchii*), and 2.43 mM for B1 (*S. minutum*), which would cause complete breakdown of PSII activity in the algae. ROS signals in type C3 treated with the ASW control (A) and menthol-treated *Symbiodinium* types B1, D1a, C15, C3, and C1 (B∼F) are presented with a representative photo; $n = 3$, scale bar in the photo represents 50 $\mu$m.

## DISCUSSION

This study indicated that millimolar levels of menthol significantly inhibited PSII activities of five freshly isolated *Symbiodinium* types. Loss of photosynthetic activity in *Symbiodinium* might explain the coral bleaching phenomenon of menthol-treated corals and sea anemones (*Wang et al., 2012a*; *Dani et al., 2016*; *Matthews et al., 2016*). However, various sensitivities
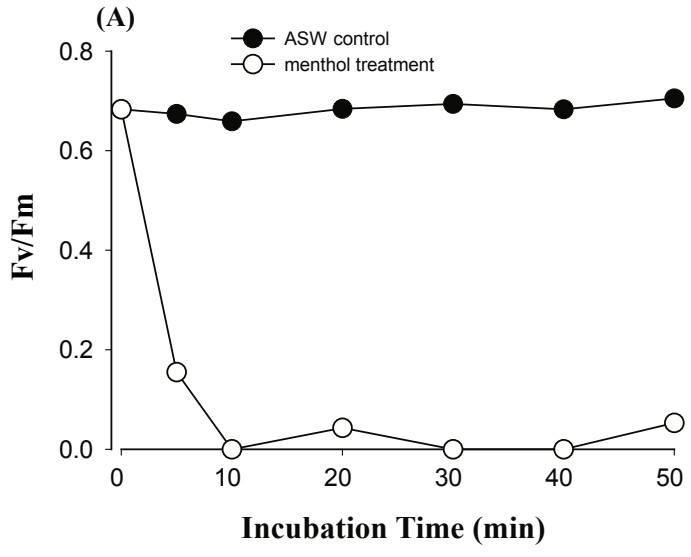

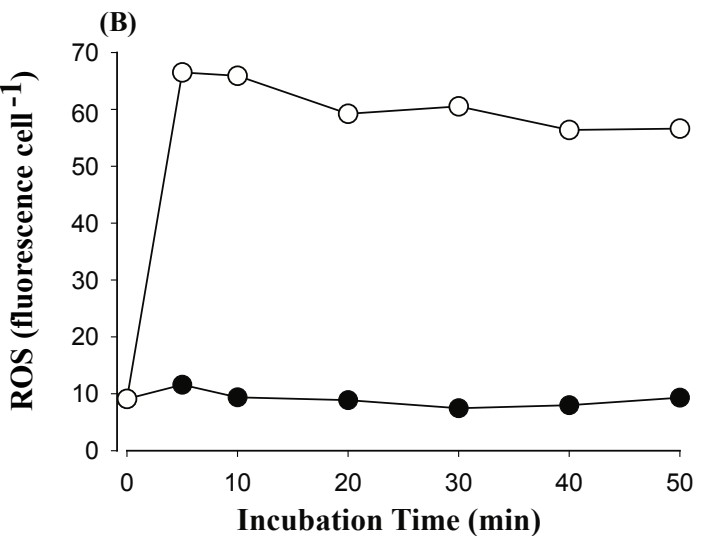

**Figure 3** **The time course of photosystem II (PSII) activity decline and reactive oxygen species (ROS) generation when treating the *Symbiodinium* C3 with menthol and artificial seawater (ASW).** Changes in PSII activity (A) and ROS levels (B) in type C3 treated with 1.73 mM menthol were determined and plotted with incubation time. ASW-incubated congeneric *Symbiodinium* was used as a control.

and response modes to menthol irritation indicate that different *Symbiodinium* types did not react uniformly to the chemicals. Diverse physiological performances among different *Symbiodinium* types were observed with respect to thermal tolerance (*Rowan, 2004*; *Tchernov et al., 2004*; *Robinson & Warner, 2006*; *Sampayo et al., 2008*; *Wang et al., 2012b*; *Suggett et al., 2015*), thermal stress-induced reactive oxygen release and antioxidant plasticity (*Krueger et al., 2014*), and expression of photosynthesis-related genes (*Parkinson et al., 2016*). Therefore, it is reasonable to expect that different *Symbiodinium* types would exhibit diverse sensitivities to menthol (Fig. 1, Table 1). However, when PSII activities were

completely shut down by menthol, responses of the *Symbiodinium* types could be divided into two groups, ROS generating (*S. goreaui*, C3, and C15) and non-ROS generating (*S. minutum* and *S. trenchii*). The reason for such a distinction needs to be addressed in future studies.

Menthol and other monoterpenes, because of their lipophilic nature, are known to inhibit growth of plant tissues by inducing lipid oxidation, which affects the membrane structure and function (*Zunino & Zygadlo, 2004*; *Singh et al., 2006*; *Kaur et al., 2011*). As proposed by *Wang et al. (2012a)*, the mechanism of menthol-induced bleaching might be attributable to $Ca^{2+}$ stimulated exocytosis. This has been demonstrated in previous results from transcriptomic studies suggesting the disruption of calcium homeostasis in both corals and sea anemones during stress-induced bleaching (*Desalvo et al., 2008*; *Moya et al., 2012*). Also, *Dani et al. (2016)* have proposed that menthol treatment could be responsible in inducing symbiophagy through $Ca^{2+}$-triggered mechanisms. Preliminary experiments conducted by *Wang et al. (2012a)* also suggest that menthol inhibition of *Symbiodinium* PSII activity may play a role in the expulsion of the algal cells or the digestion of the *Symbiodinium* cells by the host.

Generation of ROS in the menthol-treated clade C *Symbiodinium* might be derived from a lipid-oxidation process or a reaction of a TRPM8-like channel receptor. In *Symbiodinium* C3, ROS generation was initiated and reached a maximum within 5 min of incubation (Fig. 3B), indicating a typical ROS burst reaction as found in infected plants (*Wojtaszek, 1997*). The production of large amounts of ROS indicates a quick defensive response to stress. As to *S. minutum*, the lack of ROS detected in the menthol-induced PSII shutdown might be attributed to cells containing higher antioxidant activities and/or a lack of a TRPM8-like channel receptor. Comparisons of antioxidant networks revealed that *S. minutum* produced more superoxide dismutase and ascorbate peroxidase than *S. goreaui* when confronting heat stress (*Krueger et al., 2014*). No ROS generation in menthol-treated *S. trenchii* might be attributed to the "stress-tolerant" nature of members of *Symbiodinium* clade D (*Toller, Rowan & Knowlton, 2001*; *LaJeunesse et al., 2008*), in which ROS levels were found to remain unaffected by heat stress (*McGinty, Pieczonka & Mydlarz, 2012*).

In conclusion, activity of (TRP)M8 might not be a general phenomenon. We observed diverse responses based on *Symbiodinium* types to menthol treatment. In addition to (TRP)M8, menthol might also be inducing symbiophagy (*Dani et al., 2016*) in the hosts of certain *Symbiodinium* types. Although, studies (including this one) have tried to understand the mechanisms involved in loss of *Symbiodinium* cells when treated with menthol, the clarity as to what really happens is still lacking and needs to be explored in more detail in future studies.

## ACKNOWLEDGEMENTS

The authors would like to thank members of the Coral Reef Evolutionary, Ecology and Genetics (CREEG) Group, Biodiversity Research Center, Academia Sinica (BRCAS) for field support.

### Funding
This work was supported by the Ministry of Science and Technology (MOST 104-2621-B-127-001-MY3) to JTW. The funders had no role in study design, data collection and analysis, decision to publish, or preparation of the manuscript.

### Grant Disclosures
The following grant information was disclosed by the authors:
Ministry of Science and Technology: MOST 104-2621-B-127-001-MY3.

### Competing Interests
The authors declare there are no competing interests.

### Author Contributions
- Jih-Terng Wang conceived and designed the experiments, performed the experiments, analyzed the data, contributed reagents/materials/analysis tools, wrote the paper, prepared figures and/or tables, reviewed drafts of the paper.
- Shashank Keshavmurthy analyzed the data, wrote the paper, prepared figures and/or tables, reviewed drafts of the paper.
- Tzu-Ying Chu performed the experiments, analyzed the data, prepared figures and/or tables.
- Chaolun Allen Chen conceived and designed the experiments, contributed reagents/materials/analysis tools, wrote the paper, reviewed drafts of the paper.

### Field Study Permissions
The following information was supplied relating to field study approvals (i.e., approving body and any reference numbers):
The coral sample collecting permit was granted by the Kenting National Park Authority, Taiwan as part of the long-term environmental monitoring project.

### Data Availability
The raw data is supplied as a Supplemental File.

### Supplemental Information
Supplemental information for this article can be found online at http://dx.doi.org/10.7717/peerj.3843#supplemental-information.

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
