# Peer review of "Diverse responses of Symbiodinium types to menthol and DCMU treatment"

_PeerJ, doi:10.7717/peerj.3843_

## Round 0.1 · original submission · Major Revisions

I have heard back from two reviewers. Both felt your manuscript was a worthy addition to the literature. However, both reviewers have added extensive constructive comments that will require your attention, and for this reason, my decision is 'Major revisions' are needed. You will note that none of the comments appear logistically very difficult to address.

I look forward to receiving a revised manuscript.

·

Basic reporting

The introduction is too sparse. For example at L60 the authors state the mechanisms of menthol bleaching are “not clear,” but the cited references discuss the potential for menthol to interfere with the action of membrane receptor TRPM8 and lead to calcium-stimulated expulsion. These ideas are included in the discussion but they should be addressed in the introduction as well.

The authors state in L64 of the introduction: “Whether the PSII breakdown in endosymbiotic Symbiodinium in corals is directly or indirectly caused by menthol irritation of cnidarian hosts was also evaluated.” However, these results aren’t reported or discussed. To fit the data presented, this line needs to be removed, but if the authors did investigate it, I would rather they include the data.

The introduction could do a better job of establishing that no other studies have compared the responses of different Symbiodinium to both menthol and DCMU, and that this work investigates the important but underexplored topic of functional diversity among Symbiodinium species.

Across all figures and tables, please use the same order for the Symbiodinium types. Right now each figure has a different order. For Figure 2, I recommend including a label of the Symbiodinium type and menthol concentration in each panel.

Experimental design

The replication scheme should be reported in the materials and methods, not in the results.

Three of the Symbiodinium types are taxonomically described, and these names should at least be included at the first mention of each type: B1 = S. minutum, D1a = S. trenchii, C1 = S. goreaui

Validity of the findings

It appears that the ROS assay fluorescence data (presented in Table 3) was not analyzed statistically, even though replicates were measured. Such an analysis should be performed (or the reason it was not performed should be stated explicitly). Also, Table 3 should include the background control fluorescence values (reported in L167).

In the results (L155), the authors note that DCMU-treated Symbiodinium showed no ROS generation. I feel like this point should be addressed in the discussion, specifically tying back to the Jones 2004 study where the working model was that DCMU drove ROS production and subsequent bleaching.

The idea presented in L192 that clades B and D are more closely related to each other than either are to clade C is incorrect. B and C share a more recent common ancestor (take a look again at Pochon et al. 2014).

In L194-200, the authors discuss the mechanisms of menthol bleaching. As I mention earlier, this should be addressed a bit more in the introduction. In the discussion, I would like to see more explanation of how the results of this study inform our understanding of the mechanism of menthol bleaching, especially with respect to ROS production. The results run counter to previous ideas (e.g. Jones 2004) and this should be highlighted.

In L204, the authors suggest Symbiodinium minutum (B1) may lack a TRPM8 homolog. This would be easy to check, as the S. minutum genome is published and easily BLAST-able at reefgenomics.org, as are the genomes of S. microadriaticum (A1) and S. kawagutii (F1). Unfortunately, those other species were not tested here, but I think it would be valuabe to check all three anyway.

The discussion ends abruptly and does not adequately address the implications of the results. What does this mean for researchers considering using menthol or DCMU for future studies?

Additional comments

Wang et al. exposed several types of Symbiodinium to two common chemicals used in studies of coral bleaching (DCMU and menthol) to investigate their effects on photochemical efficiency and reactive oxygen species production. The authors establish functional variation among species with respect to the point at which 50% photosystem II inhibition occurs and the level of ROS that is produced.

This is useful data, but the manuscript is incomplete in several ways. The introduction is brief and presents an idea that is not tested, a key result lacks statistical analysis, and the discussion does not do a good job of exploring the implications of the findings. A major revision would be required to make the manuscript acceptable.

John Parkinson
Oregon State University

Minor Comments

Abstract
L23: change to “have been studied extensively”
L24: “cell depletion” is ambiguous. Later on it becomes clear you mean coral ‘bleaching’ (in the broadest sense, any loss of symbiont cells or reduction in photosynthetic pigmentation). I would use this term instead of “cell depletion” throughout the manuscript.
L31: as is, the sentence implies that B1 is most sensitive, while C3 and C1 are the least sensitive, when in fact the result is the opposite. Just change “PSII sensitivity” to “PSII tolerance” to fix this.
L35: could use a summary/implication statement here

Introduction
L41: change to “nine Symbiodinium clades (A-I) and numerous subcladal types”
L48: include additional examples and references for other types of functional variation (as you do in L183-187).
L48-49: rephrase or delete the clause that starts with “which provides a chance…” As written, the statement is host-centric and an oversimplification.
L54: change to “evidence has been given”
L57: change to “cells have been studied extensively” and provide references

Materials and Methods
L93: start at new paragraph at “When determining…”
L95: need to indicate that Fv/Fm values >0.6 are generally considered healthy/normal
L98: spelling. “photosynthetically”
L110: delete “Practically,”
L111: explain why values of 1.73 mM menthol and 0.13 mM DCMU were used (also, everywhere else DCMU concentration is reported in pM—why not do the same here?)

Results
L126: replication should be described in the methods. Unless it’s really important, I don’t think indicating triplicate sampling should be done so often in the results.
L160: change “significant” to “measurable” or some other word; I prefer to use ‘significant’ only in the context of statistical results.
L173: move the clause that begins with “indicating a typical ROS burst” to the discussion, talk about it more, and include references.

Discussion
L180: again, I’d avoid “algal-depletion” and use coral bleaching
L183: change to “Diverse physiological performances among different Symbiodinium types have been observed with respect to thermal tolerance…”
L206-209: Rephrase. This is a bit misleading, because D1a did produce ROS, while B1 did not (but B1 isn’t considered to be as stress-tolerant as D1a). Moreover, one of the most heat resistant Symbiodinium is S. thermophilum, a C species. This is why broad generalizations about clades should be avoided.

Reviewer 2 ·

Basic reporting

I commend the authors on a clearly written and professionally composed manuscript. If there were any areas where the text could be improved, lines 54-55 “…indicate coral bleaching caused by ROS generated by DCMU-treated Symbiodinium” reads a little unclear, and perhaps the term coral bleaching needs to be defined for the wider reader. Also Line 50 “…Symbiodinium depletion” may need to indicated that this is depletion from cnidarian host cells.

The references are generally sufficient, although I would like to suggest some more recent references in some areas:
Line 41 – there are many more detailed and recent references on the translocation of photosynthetic products and nitrogen cycling, e.g. Hillyer et al 2017, and Pernice et al 2015. A more relevant review of compound exchange than Douglas 2010 is Davy et al 2012.
Line 49 – needs a reference e.g. the review by Blackall et al 2015.
Lines 50-53 – there are many more recent papers investigating the mechanism of Symbiodinium depletion from host cells, e.g. the review by Weis 2008.
Line 80 – “Exaiptasia” needs the reference Grajales and Rodrigues, 2014.
Line 173-174 – need a reference for this statement.

The raw data are complete and the tables and figures are all necessary to better understand the results. My only comments are that sample numbers would be helpful in the figure and table legends for readers to better interpret the results.

Experimental design

The methods are well explained and utilise well-established methods of investigating Symbiodinium functionality. However, I think some justification as to why all Symbiodinium types were tested under the same 25˚C temperature is required in the text. While I appreciate this is a traditional reductionist approach (to target PSII differences in response to the chemical treatments), this decision needs to be alluded to in the text. As shown recently by Silverstein et al (2017), there are variable effects of temperature on photodamage between Symbiodinium types. The 25˚C may have been suitable to the B1 symbionts (as this is presumably, the temperature the anemone hosts were maintained under?), but this may not have been the case for the other types e.g. we know D1a photosynthetic performance is hindered under low temperatures (e.g. Silverstein et al 2017). For example, line 148-149 – are B1 the most tolerant because this is their optimal temperature? When considering Symbiodinium type responses in terms of photosystem function and ROS production, some consideration of the experimental temperature conditions will make these methods and interpretation more robust.

Other areas where the clarity of the methods could be improved are:
Line 79 – “corals were acclimated to lab conditions…” could be improved if there was a reference to another section of this manuscript or previous publication where these are detailed. The same point can be made for line 97.
Line 86 – DGGE is not currently thought to be the most effective way of identifying Symbiodinium type composition (e.g. Wham and LaJeunesse 2016), particularly if there may have been some background types, as suggested in line 86 with the description that only the “dominant” type was identified. This is particularly important in regards to the overall functionality of the Symbiodinium population, as background populations are functionally active (e.g. Cunning and Baker 2014, Cunning et al 2015).
Line 110 – A reference for how the Symbiodinium cells were counted would support these methods.

Validity of the findings

This study provides impactful information regarding the functional variability between Symbiodinium types. Specifically, this manuscript provides intriguing findings regarding the variability in sensitivity and ROS production between types. The statistics are well conducted and valid.

Additional comments

This manuscript by Wang et al provides a very important and interesting first step to understanding the mechanisms of menthol-induced Symbiodinium depletion in host cells, with vital comparisons to the traditional DCMU treatment. Furthermore, this paper is of great value to our understanding of the diverse responses between Symbiodinium types, particularly in ROS response.
Additional areas that may improve the flow of the text they are:
Lines 131-125 – The equations were very useful, although the authors may wish to consider whether they would be better placed in the methods section.
Line 194 -200 – This paragraph provides a useful background understanding of the cellular responses to menthol, and provides context for the following paragraph. For this reason, the authors may wish to consider combining the two paragraphs into one.

---

## Round 0.2 · Minor Revisions

Both reviewers have found your manuscript to be much improved, with only some small issues remaining, and thus my decision is "minor revisions" are needed.

Again, I look forward to seeing the revised version of your manuscript.

·

Basic reporting

no comment

Experimental design

no comment

Validity of the findings

no comment

Additional comments

This revision is much improved—thank you for addressing most of my concerns. While I would still like to see a statistical analysis for the data presented in Table 3 to formally test for differences among species (as it is one of the key findings in of the experiment), I realize you have a small sample size and the effect is fairly obvious. I’d personally run a few more trials to boost the numbers for an ANOVA, but I leave that choice to you.

Regarding the TRMP8 blast, I think you used a human nucleotide sequence to query the Symbiodinium nucleotide databases using a nucleotide blast (blastn). Instead, you’d want to use a protein blast (blastx). There’s likely far too much divergence between humans and Symbiodinium to get any decent nucleotide hits. Instead, blastx will translate the human nucleotide sequence to an amino acid sequence, which is more likely to be conserved across distant taxa. When I tried it there were no good hits to S. minutum, which supports the idea that TRMP8 is absent. But there weren’t any good hits to S. microadriaticum or S. kawagutii either, so this may not explain the differences in ROS production among species. It was worth checking but the results are inconclusive because you don’t have ROS data for the other two species so I’d probably leave the blast analysis out.

The rest of my comments are all minor text suggestions.

John Parkinson
Oregon State University

L38: Awkward grammar. Do you mean “This study further confirmed that Symbiodinium have diverse photochemical responses to stress even within the same clade.”

L57: “coral host” to “coral hosts”

L58: “(reviewed in Weis 2008, from cnidarian host cells, is generally attributed” to “(reviewed by Weis 2008) from cnidarian host cells is generally attributed”

L81-84: It’s a bit strange to reference a result of this study in the introduction, but ok.

L98: “clades” to “types”

L104: “Symbiodinium was isolated” to “Symbiodinium were isolated”

L118: “were identified as Symbiodinium C1, C3, C15, D1a, and B1” to “were identified as Symbiodinium ITS2 types C1 (S. goreaui), C3, C15, D1a (S. trenchii), and B1 (S. minutum)” Sorry to nitpick on how you include species names, but I really think you want to make sure to link ITS2 type to the binomial when they are first introduced in the manuscript (excluding the abstract).

L127-128: reference Fitt et al. 2001 Coral Reefs for the normal/healthy Fv/Fm value

L145: “S. minimum” to “S. minutum”

L168: delete “data indicated that”

L176: “S. minimum” to “S. minutum”

L180: delete “Table 2 indicates that”

L182: delete “A statistical analysis suggested that”

L195: “S. minimum” to “S. minutum”

L198: delete “Table 3 indicates that”

L201: consider replacing “B1” with “S. minutum” just to be consistent.

L220: “observed to thermal tolerance” to “observed with respect to thermal tolerance”

L222: “expressions” to “expression”

L240: “Symbiodinium C clade” to “clade C Symbiodinium”

L244: “huge” to “large”

L244: “defense strategy to stress” to “defensive response to stress”

L245: “no ROS generation” to “the lack of ROS generation”

L249: “the most “stress-tolerant” nature of the D clade among Symbiodinium types” to “the stress-tolerant nature of members of Symbiodinium clade D”

L255: “symbiophagy (Dani et al. 2016) in certain Symbiodinium types” to “symbiophagy (Dani et al. 2016) in the hosts of certain Symbiodinium types”

Reviewer 2 ·

Basic reporting

Generally, I am satisfied with the authors rebuttal to the concerns raised in the previous reviews. There are some spacing issues that need to be rectified (e.g. different hanging indent space, line 137), and font issues in the legend of table 2.

Lines 81-84, I am confused by this sentence – are these results from this current study? If so, it does not belong in the introduction, and if not, which study is it from?

Line 58 – missing parenthesis

Line 126 – delete one “the”

References – missing Davy et al 2012 reference

Experimental design

I am satisfied with the changes made following the previous review.

Validity of the findings

The new manuscript is improved in a number of sections here, especially if the TRPM8 homolog comparison can be included

Additional comments

This is much improved on the previous submission.

---

## Round 0.3 · accepted · Accept

The manuscript has been well revised - great work! I look forward to seeing the published version of your paper.